# The Effect of Plug Height and Inflow Rate on Water Flow Characteristics in Furrow Irrigation

Juan Yu [1,2,†], Keyao Liu [1,2,†], Anbin Li [2,3], Mingfei Yang [1,2], Xiaodong Gao [1,3,4], Xining Zhao [1,3,4] and Yaohui Cai [1,3,4,*]

1  Institute of Water-Saving Agriculture in Arid Areas of China, Northwest A&F University, Yangling 712100, China
2  College of Water Resources and Architectural Engineering, Northwest A&F University, Yangling 712100, China
3  Institute of Soil and Water Conservation, Northwest A&F University, Yangling 712100, China
4  Institute of Soil and Water Conservation, Chinese Academy of Sciences and Ministry of Water Resources, Yangling 712100, China
*  Correspondence: caiyh@nwafu.edu.cn; Tel.: +86-029-87012616
†  These authors contributed equally to this work.

**Abstract:** Despite its wide application across arid land types, furrow irrigation is often associated with numerous environmental problems related to deep percolation, runoff, and soil erosion. In this study, a straightforward approach was proposed to achieve higher uniformity and reduce erosion. Here, the impacts that a moveable "plug" has on the behavior of irrigation water in the furrow were simulated using FLOW-3D and HYDRUS-2D, where three plug heights and two flow rates were set. The effect of inflow rate and plug height on the water advance, water level, cumulative infiltration in the furrow, and uniformity coefficient was determined. Results indicate that the plug was able to slow water velocity by approximately 60% in the furrow and increase the furrow advance time by 3–4 times; the water level was increased by nearly 10 cm compared with no plug. Moreover, an irrigation uniformity range of 90.18–99.22% was associated with this plugging. The addition of a plug in the furrow irrigation practices for smallholder farmers in developing countries demonstrates great potential in reducing the probability of erosion under large slopes and can effectively improve irrigation uniformity.

**Keywords:** furrow; CFD; irrigation uniformity; cumulative infiltration





## 1. Introduction

The Global Food Crisis Report stated that in 2018, approximately 113 million people in 53 countries were still severely hungry, with all 53 countries classified as developing countries [1]. Conflict, climate change, and economic recession remain the main causes of the food crisis. In particular, climate change and population growth have increased water scarcity in these areas [2]. The current growing global food demand and increasingly fierce competition between crop and environmental water have resulted in severe complications in determining how to effectively increase crop yields, reduce agricultural water use, and improve agricultural water efficiency for developing countries [3,4].

In developing countries, low adoption of irrigation technology is widespread, particularly relating to drip and sprinkler irrigation [5]. At present, in the areas of extreme water shortage in Africa and Asia, traditional surface irrigation methods are still being used on a large scale [6–8]. Therefore, improvements in surface irrigation technology will increase farmer production and income in these areas and consequently aid in food crisis and water shortage alleviation.

Furrow irrigation is widely used due to its inherent advantages, including low capital costs, minimal energy requirements, and simple operation [9]. However, furrow irrigation

is not applicable in areas with large slopes and sandy soil, due to the possibility of serious deep percolation and surface erosion [10]. How to realize the rational use of water through the optimization of the furrow irrigation system has become an urgent problem to be solved [11]. Sayari et al. [12] identified a lower inflow rate and appropriate irrigation time to improve management outcomes in furrow irrigation systems, reducing water use compared to other surface irrigation methods, and water resources can be used effectively when the furrow irrigation methods are designed properly. Mohammadi et al. [13] demonstrated increases in grain yield and irrigation water productivity of 50% and 67%, respectively, due to the simultaneous management of fertilization, irrigation, and plant density.

Surge flow irrigation technology is based on furrow irrigation, with the aim of improving surface irrigation. This irrigation approach has attracted widespread attention from scholars due to its effective water and energy saving abilities and fertilizer conservation, amongst other benefits [14]. During the surge flow irrigation process, the surface is repeatedly wetted and dried, resulting in the formation of a dense layer that directly affects surface infiltration capacity. This layer can save up to 10–40% of water compared to furrow irrigation under the same conditions [15]. However, surge flow irrigation requires a surge valve, controller, and water distribution pipeline, etc., and is thus more expensive than furrow irrigation. A simple and feasible method is therefore urgently required in order to improve the irrigation efficiency and uniformity of furrow and surge flow irrigation [16].

Jiang et al. [17] analyzed the uniformity of cotton growth indicators and yield along the irrigation ditch under furrow irrigation conditions, and the furrow irrigation water and nitrogen management measures, such as irrigation amount and nitrogen application, have a significant impact on the uniformity of cotton growth and yield along the irrigation ditch. Wu et al. [18] evaluated the uniformity of irrigation distribution by optimizing the appropriate combination of alternate furrow irrigation (AFI) and conventional furrow irrigation (CFI) irrigation technical parameters. Mulubrehan et al. [19] studied the significant effect of the irrigation flow method on irrigation application efficiency and distribution uniformity. Over recent years, farmers in some regions of China have spontaneously invented a simple method to improve furrow irrigation. More specifically, a simple plug (e.g., straw or bags) is employed to move with the water in the furrows (Figure 1). This subsequently increases the water level in the furrows under a small inlet flow in order to achieve a higher cumulative infiltration. However, the ability of this method to save water and improve uniformity remains unclear.

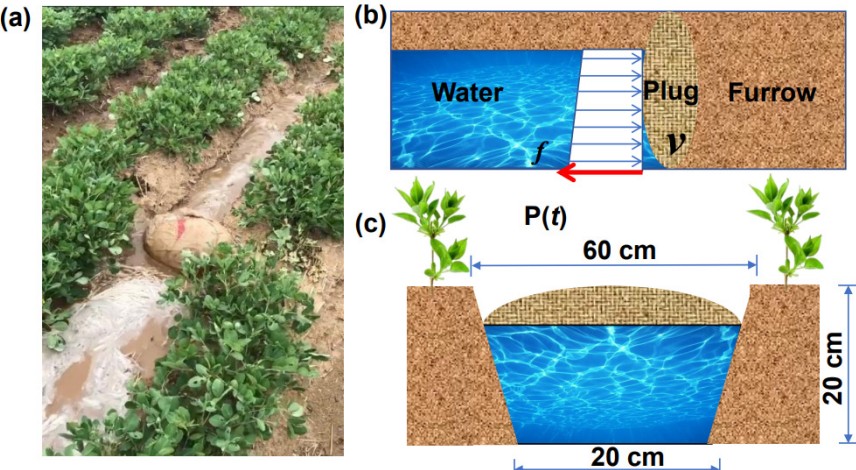

**Figure 1.** Overview of the furrow and plug. (**a**) Photo from the field. (**b**) Longitudinal section of the furrow. (**c**) Cross section of the furrow.

Moreover, experimental works are labor intensive, and experimental results are influenced by the model size, measurement accuracy, etc. Therefore, the application of theoretical analysis and numerical simulation methods has become a practical approach

to analyzing furrow irrigation characteristics [20]. Despite the availability of numerous software systems to analyze the performance of furrow irrigation systems (e.g., WinSRFR 3.1 and HYDRUS 3.02), the majority are unable to accurately simulate the plug in the furrow and the subsequent processes [11,21]. FLOW-3D is highly-efficient and offers a comprehensive solution for free-surface flow problems. In recent years, FLOW-3D has been employed to solve multiple problems in irrigation, including the hydraulic characteristics of U-shaped channels and Labyrinth weirs [22,23]. However, FLOW-3D fails to fully consider the infiltration characteristics of furrow irrigation. HYDRUS-2D is the most widely used computer software package for the prediction of water movement in two-dimensional, variably saturated porous media. Numerous studies have simulated water movement under furrow irrigation using this model [24–26]. Qiu et al. [27] used HYDRUS-2D to simulate the soil moisture dynamics of greenhouse tomatoes under furrow irrigation, and they found that the HYDRUS-2D model with consideration of root water uptake can be used to improve irrigation scheduling for furrow irrigated tomato plants in greenhouses in arid regions. However, the dynamic processes of water advancement, storage, and infiltration during a furrow irrigation event could not be described by just using the FLOW-3D and HYDRUS-2D models, especially if a plug was added to the furrow.

In the current study, FLOW-3D and HYDRUS-2D were combined to investigate the effect of the inflow rate and plug height on the plug operating speed, water level, and cumulative infiltration at different positions within the furrow. Furthermore, the irrigation quality of furrow irrigation was evaluated via the uniformity coefficient.

## 2. Materials and Methods

### 2.1. FLOW-3D Simulation

#### 2.1.1. Governing Equations

The FLOW-3D Re-normalization group k-$\varepsilon$ model (RNG), based on the RANS equation, is a turbulence model suitable for turbulent flow fields with a high strain rate and large streamline curvature at the front of the plug. The continuity and RANS equations for an incompressible, viscous flow in the Cartesian coordinate system are given by Equations (1) and (2): 

Continuity equation:

$$\frac{\partial \rho}{\partial t} + \frac{\partial u_i}{\partial x_i} = 0 \tag{1}$$

RANS equation:

$$\frac{\partial \rho \, \overline{u_i}}{\partial t} + \frac{\partial \overline{u_i u_j}}{\partial x_i} = \mu \frac{\partial}{\partial x_j} \left( \frac{\partial \, \overline{u_i}}{\partial x_j} \right) - \frac{\partial \, \overline{p}}{\partial x_i} + \frac{\partial \left( -\rho \overline{u_i u_j} \right)}{\partial x_j} \tag{2}$$

where $\rho$ is the fluid density, u is the velocity, p is the pressure, t is the time, $\mu$ is the dynamic viscosity of the fluid, and $-\rho \overline{u_i u_j}$ is the Reynolds stress.

Furthermore, the k (turbulence kinetic energy) and $\varepsilon$ (dissipation rate) equations are described using Equations (3) and (4):

k equation:

$$\frac{\partial \rho k}{\partial t} + \frac{\partial \rho k u_i}{\partial x_i} = \mu \frac{\partial}{\partial x_j} \left( \alpha_k \mu_{eff} \frac{\partial k}{\partial x_j} \right) + G_k + \rho_\varepsilon \tag{3}$$

$\varepsilon$ equation:

$$\frac{\partial \rho \varepsilon}{\partial t} + \frac{\partial \rho \varepsilon u_i}{\partial x_i} = \frac{\partial}{\partial x_j} \left( \alpha_\varepsilon \mu_{eff} \frac{\partial \varepsilon}{\partial x_j} \right) + \frac{C_{1\varepsilon}^* \varepsilon}{k} - C_{2g} \rho \frac{\varepsilon^2}{k} \tag{4}$$

where $\mu_{eff} = \mu + \mu_t$ is the diffusion coefficient, $\alpha_k$ and $\alpha_\varepsilon$ are constants and $\alpha_k = \alpha_\varepsilon = 1.39$, $C_{2\varepsilon} = 1.68$ is the model constant with $C_{1\varepsilon}^* = C_{1\varepsilon} - \frac{\eta(1-\eta/\eta_0)}{1+\beta\eta^3}$ for constants $C_{1\varepsilon} = 1.42$, $\eta_0 = 4.377$, $\beta = 0.012$, $\eta = (2E_{ij} \cdot E_{ij})^{1/2} \frac{k}{\varepsilon}$, $E_{ij} = \frac{1}{2}\left(\frac{\partial u_i}{\partial x_i} + \frac{\partial u_j}{\partial x_i}\right)$ is the time average strain rate and $\mu_t = \rho C_\mu \frac{K^2}{\varepsilon}$ is the eddy viscosity coefficient, with $C_\mu = 0.0845$.

### 2.1.2. Grid Division

In order to minimize the temporal costs of the iteration, a furrow model with a length of 30 m was selected. Figure 2 demonstrates the local computational domain and furrow size of the numerical model, with the total number of grids set as 550,000.

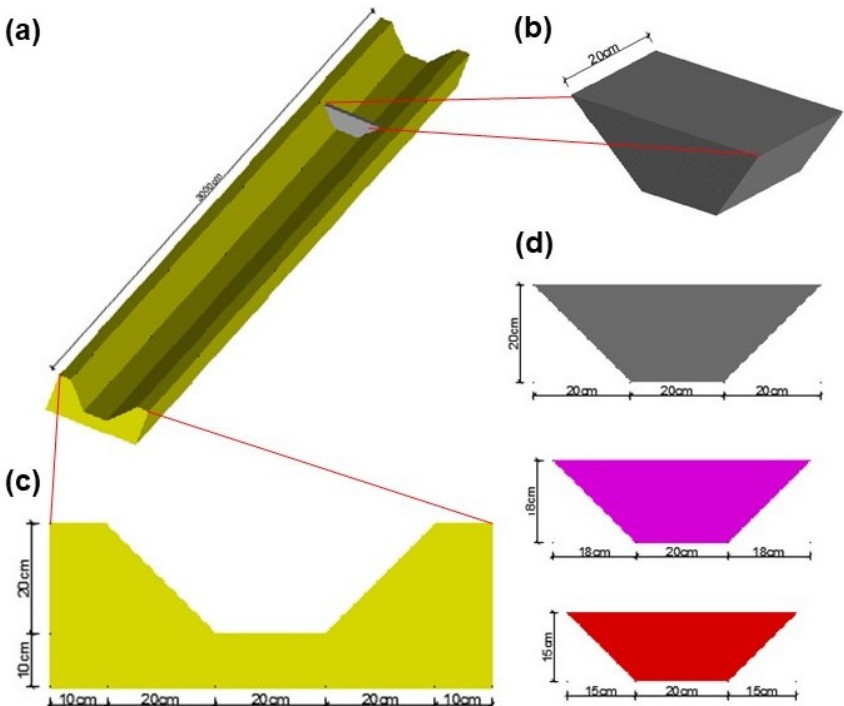

**Figure 2.** General view of the furrow and plug for the simulated furrow irrigation scenario. (**a**) Simulated furrow. (**b**) Plug. (**c**) Cross-sectional size of furrow. (**d**) Cross-sectional size of plug.

### 2.1.3. Boundary Conditions

The boundary conditions, based on the actual situation, are described as follows. The water flow in the simulated furrow was constant; the upstream inlet boundary of the furrow was set to the "volume flow rate" option; the downstream outlet boundary was set to the "outflow" option; the furrow and bottom wall boundaries were set to the "wall" option; and the upper region of the furrow was set as the "specified pressure" option, whose pressure is equal to the atmospheric pressure.

### 2.1.4. Working Conditions

Simulations were performed using FLOW-3D (v10.1.1), with the GMO (General Moving Objects) model used to simulate the interaction between the moving plug and water flow. Eight types of working conditions were set for the simulations based on the different water flow rates and plug heights. The plug is composed of a mixture of straw and soil, and the density of the plug was measured to be 1.3 g/cm$^3$ (Table 1).

**Table 1.** Simulation treatments and working conditions.

| Treatments | Flow Rate (L/s) | Field Slope | Height of Plug (cm) |
|:---:|:---:|:---:|:---:|
| A1 | 2.8 | 0.002 | 15 |
| A2 | 2.8 | 0.002 | 18 |
| A3 | 2.8 | 0.002 | 21 |
| A4 | 2.8 | 0.002 | 0 (no plug) |
| B1 | 3.0 | 0.002 | 15 |
| B2 | 3.0 | 0.002 | 18 |
| B3 | 3.0 | 0.002 | 21 |
| B4 | 3.0 | 0.002 | 0 (no plug) |

*2.2. HYDRUS-2D Simulation*

2.2.1. Governing Equations

The movement of the soil water from the furrows into the soil was simulated using HYDRUS-2D (version 2.03). For homogeneous and isotropic soil, the governing two-dimensional flow equation is described by the Richards equation, which is solved using the Galerkin finite-element method described as follows:

$$\frac{\partial \theta}{\partial t} = \frac{\partial}{\partial x}\left[K(h)\frac{\partial h}{\partial x}\right] + \frac{\partial}{\partial z}\left[K(h)\frac{\partial h}{\partial z}\right] + \frac{\partial K(h)}{\partial z} \tag{5}$$

where $\theta$ is the volumetric water content, t is the time, x and z are the horizontal and vertical coordinates, respectively, with a positive upward direction of the latter, h is the pressure head, and $K(h)$ is the unsaturated hydraulic conductivity.

The unsaturated hydraulic conductivity function is determined using the van Genuchten–Mualem model [28,29].

$$S_e = \frac{\theta(h)-\theta_r}{\theta_s-\theta_r} = \frac{1}{\left(1+|\alpha h|^n\right)^m}\,(m = 1 - 1/n) \tag{6}$$

$$K(h) = K_s S_e^{0.5}\left[1-\left(1 - S_e^{1/m}\right)^m\right]^2 \tag{7}$$

where $S_e$ is the relative saturation; $K_s$ is the saturated hydraulic conductivity; $\theta_r$ and $\theta_s$ are the residual and saturated water contents, respectively; $\alpha$ is an empirical parameter inversely related to the air entry value; and n and m are van Genuchten–Mualem shape parameters.

2.2.2. Boundary Conditions and Initial Conditions

Figure 3 presents the geometry and boundary conditions used to define the physical problem investigated in this study. Due to the symmetry of the furrows, only the right side is studied. Surface AGFE, which denotes the surface of the furrow, is set as a "variable head boundary condition", with the variable head value determined via FLOW-3D. The left and right boundaries, AB and DC respectively, were set to the "zero flux" condition, while the bottom boundary, BC, was set as a "free drainage boundary". The "atmospheric boundary condition" was selected for FED, while the water evaporation rate was set as 0.4 cm d$^{-1}$. Daily variations in the evaporation rate were not considered.

Two types of soils were used in the simulation: a typical sandy loam and loam [30]. Table 2 reports the hydraulic properties of these soils. The initial soil water content was set as 0.07 and 0.08 cm$^3$ cm$^{-3}$, respectively.

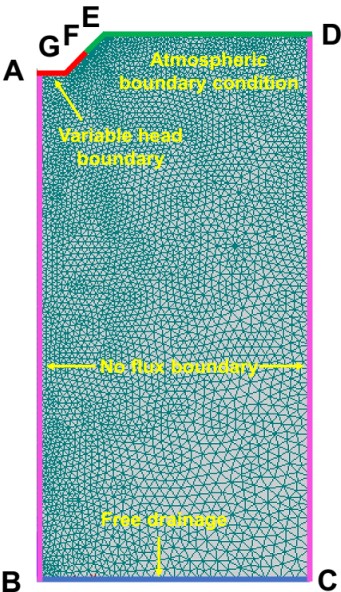

**Figure 3.** Geometric and boundary conditions of this study.

**Table 2.** Hydraulic properties of sandy loam and loam.

| Soil | $\theta_r$ (cm$^3$ cm$^{-3}$) | $\theta_s$ (cm$^3$ cm$^{-3}$) | $\alpha$ (m$^{-1}$) | n (-) | Ks (cm h$^{-1}$) |
|---|---|---|---|---|---|
| Sandy loam | 0.065 | 0.41 | 0.075 | 1.89 | 4.42 |
| Loam | 0.078 | 0.43 | 0.036 | 1.56 | 1.04 |

### 2.3. Irrigation Uniformity

The Christiansen uniformity coefficient CU quantifies the uniformity distributed along the furrow [31] and can be calculated as follows:

$$CU = \left[ 1 - \frac{\sum_{i=1}^{n} \left| Z_i - \bar{Z} \right|}{n \, \bar{Z}} \right] \qquad (8)$$

where $Z_i$ is the ith cumulative infiltration of the observed point in the furrow and $\bar{Z}$ is the average cumulative infiltration of the total n observed points.

### 2.4. Cumulative Infiltration

The infiltration of furrow irrigation can be described by the Green–Ampt model as follows:

$$i = K_s \frac{z_f + s_f + H}{z_f} \qquad (9)$$

where i is the infiltration rate; $K_s$ is saturated hydraulic conductivity; $z_f$ is the generalized wetting front depth; $s_f$ is the water suction at the wetting front; and H is the water level.

Thus, cumulative infiltration I can be determined from the integral in Equation (10):

$$I = \int_{t_0}^{t_1} i = \int_{t_0}^{t_1} K_s \frac{z_f + s_f + H}{z_f} \qquad (10)$$

## 3. Results and Discussion

### 3.1. Water Advancement in the Furrow and Plug Operating Speed

Figure 4 depicts the water advancement across time for different treatments. The presence of the plug was observed to increase the advancement time, with the higher the position of the plug, the longer the advancement time. A resistance estimate was made for the furrow irrigation channel [32]. The higher the plug, the bigger the quality of the plug, and the more energy required to drive the plug. For example, at the inflow rate of 2.8 L/s, the plug height increases from 15 cm to 21 cm, and the time taken for the water to advance to the end of the furrow increases by approximately 20 s. The longest advancement times corresponded to treatments A3 and B4, respectively.

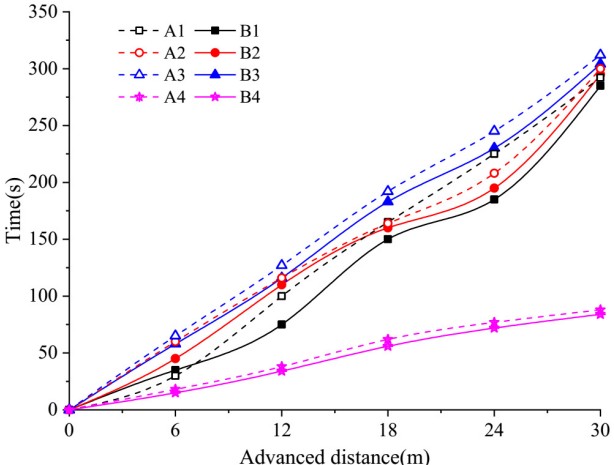

**Figure 4.** Advancement curves for all treatments.

Figure 5 depicts the simulated water flow velocity under the different treatments. Treatments A4 and B4 exhibit average velocities of 32.27 cm/s and 34.49 cm/s, respectively, with the velocities of the remaining treatments not reaching more than 8–10 cm/s. This indicates that adding a plug in the furrow can slow down the water flow speed, consequently avoiding the occurrence of scour in the furrow. Furthermore, the smaller the plug, the larger the inflow rate, and the larger flow velocity. For example, at the inflow rate of 3 L/s, a plug height increase from 15 cm to 21 cm corresponds to a decrease in velocity by 19.03%.

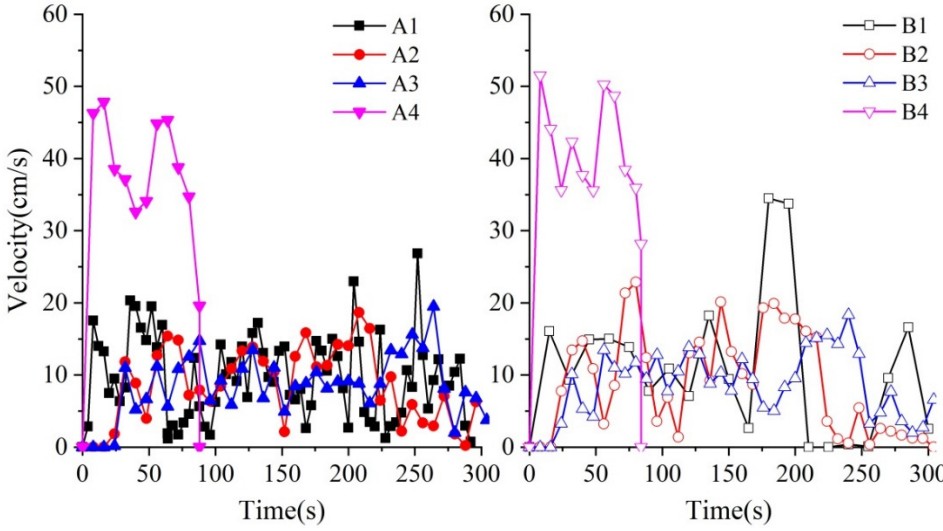

**Figure 5.** Water flow velocity under different treatments.

When a plug is added to the furrow, the water velocity is equal to the plug speed, indicating the constant fluctuations of the latter. At the inflow rate of 3 L/s and plug height of 15 cm, the speed increases rapidly after the plug moves to 160 s. This reduces the water level and consequently, the speed drops quickly to 0 m/s and is maintained at this level until the water level rises again and the plug continues moving forward. Figure 1b presents a schematic diagram of the force analysis for the plug. During operation, the plug is subjected to the water flow pressure in the furrow, as well as the friction between the plug and the furrow, generating the speed of motion. The plug speed and mass demonstrate a relationship with the inflow rate during the irrigation process. In general, the inflow rate increases with the water flow pressure, and the plug speed is faster, while the plug mass increases with the friction force, reducing the speed. The addition of the plug is associated with a decrease in flow velocity by approximately 60%. Dibal et al. [33] studied the effects of soil permeability, land slope, and furrow irrigation characteristics on furrow irrigation-induced erosion, and pointed out that the greater the slope, the more serious the erosion. The slope is the most significant factor affecting erosion in surface irrigation [34]. For furrows with larger slopes, this method can effectively prevent the scouring of the bottom of the furrow resulting from excessive water flow speed, ensuring the stability of the surface soil particles. Thus, as well as areas with larger slopes, this method is also suitable for locations with a sandy soil texture.

### 3.2. Water Levels

Figure 6 demonstrates the simulated water level when the water flows advance to 12 m, 18 m, and 30 m under the different treatments. For A4 and B4, initially the water level is 0 cm and subsequently increases with the advancement of the water flow. However, this increase does not exceed 5 cm until the water flow advances to the end of the furrow. When a plug is added to the furrow, given an initial push, the flow is able to reach a higher water level. For example, for B2 (3 L/s inflow rate and 18 cm plug height), pushing the water flow 12 m allows the water level to reach 10.54 cm. Moreover, the water advancement time and surface profile increase as the inflow rate decreases and the plug height increases.

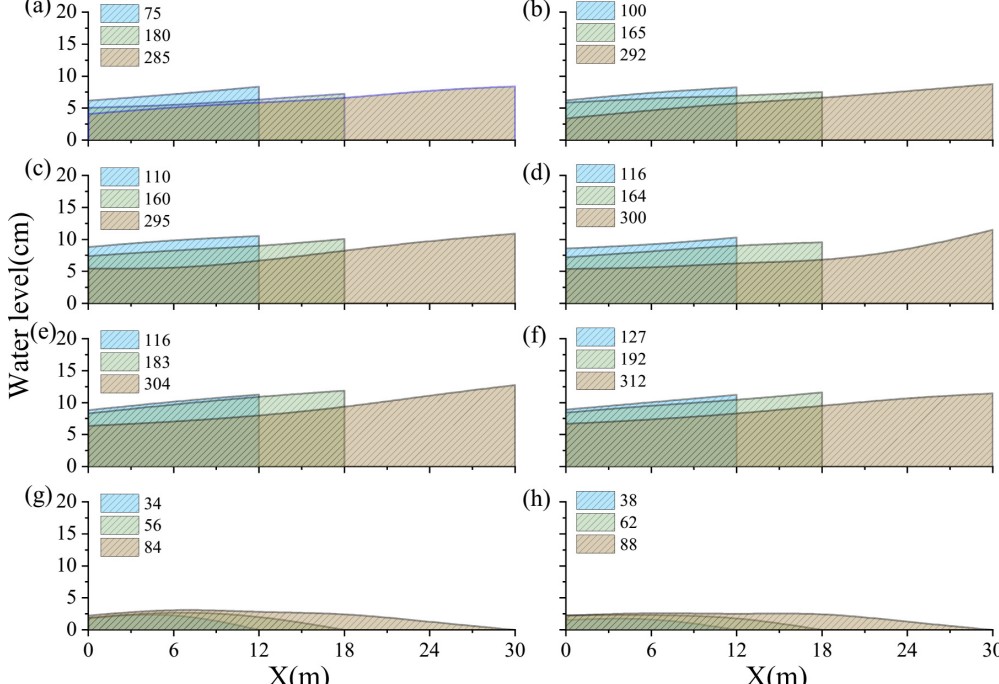

**Figure 6.** Water level for initial irrigation water positions of 6 m, 18 m, and 30 m in the furrow of different treatments. (**a**) B1; (**b**) A1; (**c**) B2; (**d**) A2; (**e**) B3; (**f**) A3; (**g**) B4; (**h**) A4.

### 3.3. Cumulative Infiltration

Although cumulative infiltration of loam is lower than that of sandy loam, the general infiltration properties of the two soils are similar, and thus for simplicity, we only present the infiltration curve for a loam in this section (Figure 7). The highest cumulative infiltration was observed for B4 at 30 m. However, the cumulative infiltration for this treatment generally exhibited great variation at each position due to the high flow velocity in the absence of plugging. The other furrow positions of B4 are associated with a relatively low cumulative infiltration, with the minimum value occurring at 0 m and differences reaching 73.97 cm$^2$. Adding a plug dramatically affected the furrow-long infiltration. The changes in cumulative infiltration are similar to the plugging treatments. The longer time required for the irrigation water to reach the end of the furrow and the large infiltration rate results in a small difference in the cumulative infiltration at each point in the furrow. The cumulative infiltration is observed to be higher for the large inflow rate treatments and increases with plug height. For example, the cumulative infiltration of B2 is 1.08 times that of A2 at the 30 m position.

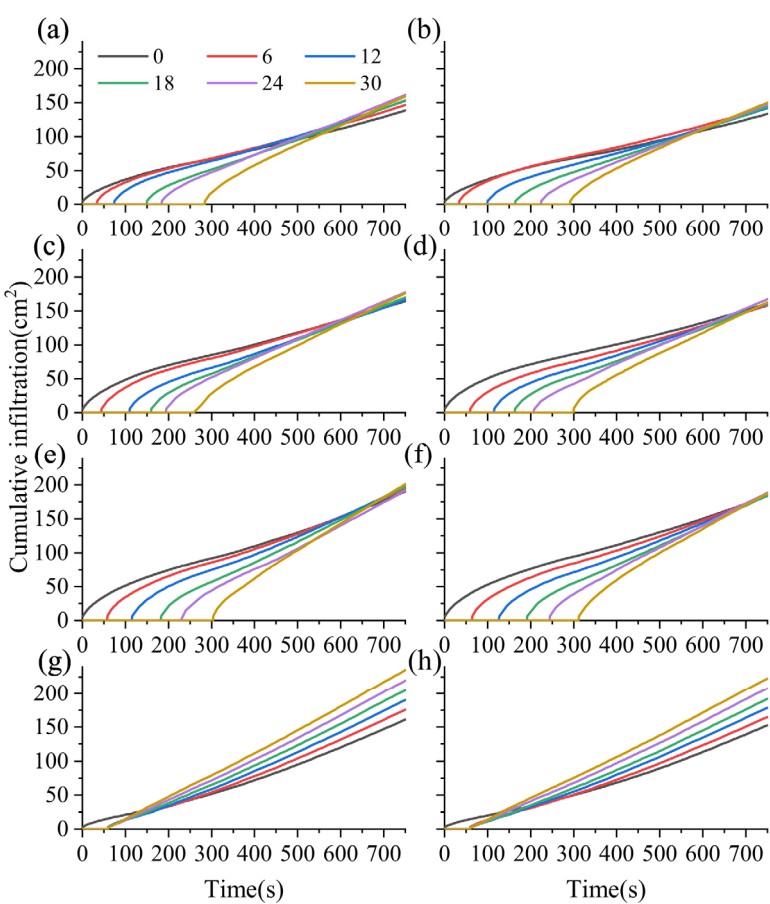

**Figure 7.** Cumulative infiltration curve for all treatments. (**a**) B1; (**b**) A1; (**c**) B2; (**d**) A2; (**e**) B3; (**f**) A3; (**g**) B4; (**h**) A4.

Variations in cumulative infiltration are generally a result of differences in water level. As the infiltration head increases, the infiltration volume increases gradually [35]. Figure 8 demonstrates the change in water level for treatments B2 and B4 for different furrow positions. With the addition of the plug, the initial water level is relatively low at 0 m and gradually decreases with time. The water level subsequently rises once the plug reaches the end of the furrow. The other sections in the B2 furrow exhibit slightly longer water flow advancements, with an evidently higher water level. Furthermore, the closer the section is to the furrow end, the higher the water level. However, B4 exhibits an extremely fast water flow speed, with the water level increasing rapidly at each position. Once the water flow

reaches the end of the furrow, the rising rate of water level significantly changes. The closer to the end of the furrow, the higher the rising rate of the water level. The greatest rising rate of the water level is observed at 30 m and is approximately equal to 20 cm, with the 0 m water level reaching just 14 cm. This indicates a large difference in cumulative infiltration resulting from water level variations.

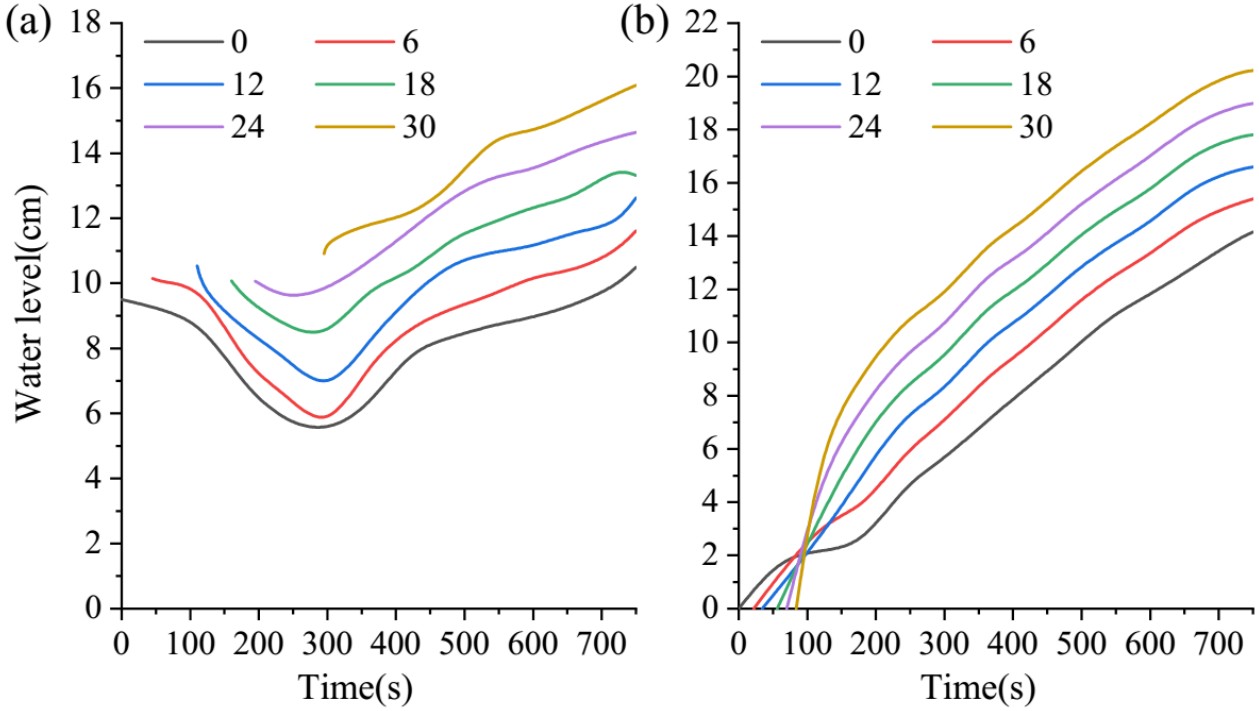

**Figure 8.** Changes in water level across different positions for (**a**) B2 and (**b**) B4.

Although the Green–Ampt model is used to study the infiltration of the initially dry soil in the thin layer of water, the water level during the infiltration process does not change. In the current study, the water level changes continuously, yet the effect of the water level on the infiltration amount can be determined via Equation (10). In particular, H increases with the cumulative infiltration of the furrow irrigation. The plugging treatments exhibit a remarkable increase in water level H at the end of the furrow, and thus the cumulative infiltration at the end of the furrow is able to reach a level equal to the head in a relatively short time. This suggests improved irrigation uniformity.

*3.4. Uniformity Coefficient*

Table 3 reports the uniformity coefficient values determined from the simulations. The irrigation uniformity is observed to range from 90.18% to 99.22% with the addition of plugging for both soils. This indicates a highly uniform distribution of the irrigation water along the furrow. For example, treatments with an inflow rate of 2.8 L/s reach an irrigation uniformity of 99.22% in sandy loam. However, with the absence of the plug, the uniformity is just 88.72–89.58%, with the minimum uniformity coefficient observed for A4 (2.8 L/s inflow rate and no plug) in sandy loam. This may be attributed to the rapid saturation of the surface soil at low water levels in the absence of a plug. This results in the soil hydraulic conductivity reaching the saturated hydraulic conductivity, and the corresponding infiltration rate decreases rapidly at the head of the furrow (0 m, 6 m), while the cumulative infiltration changes gradually. Moreover, the high-water level increase at the end is maintained, with a larger cumulative infiltration and consequently, a lower uniformity. However, the addition of the plug results in a much lower flow velocity, increasing the water level at each position in the furrow. The end of the furrow is associated with a reduction in the infiltration time and an increase in the water level, thus balancing

the cumulative infiltration at each position. Therefore, variations in cumulative infiltration are minimal for each section, which consequently enhances the uniformity.

**Table 3.** Irrigation uniformity under different treatments.

| Soil Texture | Irrigation Uniformity | | | | | | | |
|---|---|---|---|---|---|---|---|---|
| | A1 | A2 | A3 | A4 | B1 | B2 | B3 | B4 |
| Sandy loam | 97.30% | 98.60% | 99.22% | 88.72% | 95.81% | 97.49% | 98.36% | 88.84% |
| Loam | 90.18% | 96.02% | 96.01% | 89.44% | 93.11% | 94.97% | 96.72% | 89.58% |

The lower inflow rate of sandy loam increases the uniformity due to the larger saturated hydraulic conductivity of this soil type. The water level in the furrow increases with the inflow rate, resulting in a larger infiltration rate, and a slightly lower uniformity. Furthermore, the saturated water conductivity of loam is low, and thus under large inflow rates, the water level in the furrow is relatively stable, with uniform infiltration. Our results generally indicate improvement of the irrigation uniformity following the addition of the plug in the furrow.

The maximum uniformity for sandy loam is observed for an inflow rate of 2.8 L/s and a plug height of 21 cm. The corresponding values for loam are an inflow rate of 3 L/s and a plug height of 21 cm. However, a large inflow rate can cause the irrigation water to overflow over the furrow ridge, and thus a plug height of 20 cm may be required in the loam. Similar results were also found by Mazarei et al. [11], who used WinSRFR in the fields to optimize the performance of furrow irrigation, and found that the maximum performance in sugarcane fields would be obtained when the inflow rate was 3 L/s and cut-off time was 379.5 min.

In recent years, various simple methods are used to improve the irrigation quality of furrow irrigation. Sayari et al. [12] indicated meandering furrow could reduce the runoff, erosion losses, the mass of fertilizer lost, and surface water contamination by decreasing the velocity of water advance. Keshavarz et al. [36] proposed a new approach to improve water flow characteristics by creating micro-dams, barriers inside the irrigated furrows, and they found micro-dams increased the advance and recession time, leading to increased infiltration, and the uniformity coefficient ranged from 93.6 to 99.3%. The method proposed in our study can be regarded as an improvement of the method, which was conducted by Keshavarz et al. [36]. These methods have made some progress in improving the quality of furrow irrigation. More importantly, in developing countries, farmers' income is low. With these simple methods, not only does the cost of irrigation not increase but also a higher irrigation quality can be obtained. Therefore, they are beneficial to alleviating the global food crisis.

## 4. Conclusions

In the current study, we proposed a new method to improve the uniformity of furrow irrigation by adding plug-in furrows. We applied the FLOW-3D and HYDRUS-2D simulation methods in order to explore the effect of the furrow flow rate and plug height on the water advancement, water level, cumulative infiltration, and uniformity coefficient of furrow irrigation.

(1) The addition of a plug can slow the water velocity in the furrow and prevent the rapid scouring of the furrow. At an inflow rate of 3 L/s, the increase in plug height from 15 cm to 21 cm corresponds to a 19.03% decrease in velocity. The lower the inflow rate, the larger the plug height, and the slower the water velocity;

(2) The addition of plugs is able to effectively increase the trenching time and water level, thereby increasing the cumulative infiltration and improving the uniformity of irrigation. Irrigation uniformity with plugging ranges from 90.18% to 99.22%.

The inclusion of a plug in the furrow irrigation systems effectively reduces the probability of erosion under large slopes and improves the irrigation uniformity for smallholder

farmers in developing countries. Further research on this topic could focus on the actual application of this simple technology in the field, and analyze the combined effect of field slope, soil texture, and plug characteristics on the hydraulic characteristics of furrow irrigation.

**Author Contributions:** Conceptualization, Y.C., X.G. and X.Z.; methodology, Y.C., X.G. and X.Z.; formal analysis, Y.C. and A.L.; writing—original draft preparation, J.Y., M.Y. and K.L.; writing—review and editing, Y.C.; project administration, Y.C.; funding acquisition, Y.C. All authors have read and agreed to the published version of the manuscript.

**Funding:** The authors gratefully acknowledge the National Key R&D Program of China (2021YFD1900700), the National Natural Science Foundation of China (52009113, 52069016), the Key Research and Development Program of Shaanxi Province (2021NY-167, 2020ZDLNY07-04), the Young Talents Support Program of the Science and Technology Association of Shaanxi Provincial Universities (No. 20210412), CAS "Youth Scholar of West China" Program (XAB2019B09), and the Cyrus Tang Foundation.

**Data Availability Statement:** Data are presented in the article.

**Conflicts of Interest:** The authors declare that they have no known competing financial interest or personal relationship that could have appeared to influence the work reported in this paper.

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
