# Peer review of "The Effect of Plug Height and Inflow Rate on Water Flow Characteristics in Furrow Irrigation"

_agronomy, doi:10.3390/agronomy12092225_

Round 1
Reviewer 1 Report
This is a practical study, aiming to test the performance and effectiveness of a water structure that is already applied by farmers in China. Performing numerical analyses to support the experience is found to be quite interesting. The manuscript should be improved as follows before publication:
1- Literature review is quite weak. Please improve this manuscript by including more literature study on the related topics and discuss these studies to accommodate literature gap and originality of your work. Please include the studies on analyses of the performance similar to the structure that is designed within this study. The discussions may be around this topic, not purely the use of numerical algorithms.
2-The conclusion section is quite weak and there are several points that is not suitable for a scientific writing. Please improve the conclusion to better reflect and summarize the findings of this study. Do not use unsuitable words and phrases such as in lines 338 etc.
Author Response
This is a practical study, aiming to test the performance and effectiveness of a water structure that is already applied by farmers in China. Performing numerical analyses to support the experience is found to be quite interesting. The manuscript should be improved as follows before publication:
We are very excited to have been given the opportunity to revise our manuscript. We very much appreciate the time and effort you’ve put into the comments. We will keep working hard.
1- Literature review is quite weak. Please improve this manuscript by including more literature study on the related topics and discuss these studies to accommodate literature gap and originality of your work. Please include the studies on analyses of the performance similar to the structure that is designed within this study. The discussions may be around this topic, not purely the use of numerical algorithms.
Response:We agree. We have thought through your suggestions and read more literature on the subject. References 1-3 have been added to the Introduction to accommodate the gaps in the literature and the originality of our work. Added measures to improve the uniformity of furrow irrigation, instead of just emphasizing the use of FLOW-3D and HYDRUS-2D.
2-The conclusion section is quite weak and there are several points that is not suitable for a scientific writing. Please improve the conclusion to better reflect and summarize the findings of this study. Do not use unsuitable words and phrases such as in lines 338 etc.
Response: We agree. After careful reading of your suggestions, we have rephrased the conclusions to make them suitable for scientific writing and to better reflect and summarize the findings of this study.
Reviewer 2 Report
The work has merit because it addresses an interesting strategy to increase the uniformity of furrow irrigation.
However, some points should be addressed better:
1 - Measuring irrigation efficiency with the use of the plug, wouldn't it be more interesting than uniformity?
2 - The constitution of the Plug used is essential for a better knowledge of the work.
3 - Adequate methodology, please review the appropriate location for equations 9 and 10.
4 - More information in the manuscript.

Author Response
The work has merit because it addresses an interesting strategy to increase the uniformity of furrow irrigation. However, some points should be addressed better.
Response: We thank the reviewer for your valuable comments. The reviewer has a good understanding of the goal of our manuscript. We appreciate the reviewer’s constructive suggestion which will help to avoid confusion from the readers.
1 - Measuring irrigation efficiency with the use of the plug, wouldn't it be more interesting than uniformity?
Response: Thank you for your valuable suggestions. Irrigation efficiency is indeed one of the most important indicators to measure the efficiency of irrigation technology, and it can effectively characterize the water conversion efficiency under special circumstances. Therefore, we combine it with the water conversion process in another article. Thank you again for such valuable advice.
2 - The constitution of the Plug used is essential for a better knowledge of the work.
Response: We agree. We set three plug heights and two flow rates according to the actual situation.
3 - Adequate methodology, please review the appropriate location for equations 9 and 10.
Response: We agree. We have put Equations 9 and 10 into the Materials Methods.
4 - More information in the manuscript.
Response: We list the questions in the manuscript and answer them, please see the answers blow.
4.1 L16-17 It would be more interesting to describe the test (flow and plug height) proposed in the work.
Response: We agree. We set three plug heights and two flow rates according to the actual situation.
4.2 L21-22 Why not evaluate application efficiency and water requeriment efficiency?
Response: We agree. This question is the same as the first one, and we study it in a separate article.
4.3 L35-36 consumption or use?
Response: we agree, we have changed "consumption" to "use" in the article.
4.4 L79-99 should have greater emphasis on measures to increase uniformity in the furrows.
Response: We agree. We have supplemented the introduction with measures to improve the uniformity of furrow irrigation.
4.5 L149 What is the material constitution of this plug? This description is important.
Response: We agree. The stopper is constructed of a mix of straw and soil.
4.6 L183 Evaluation of application efficiency would be very important.
Response: We agree. This problem is the same as the first one, and we study it in a separate article.
4.7 L190-207 There is a lack of discussion in the work based on the literature. For example: Couldn't the high infiltration promoted by a longer advance time increase the percolation at the beginning of the furrow?
Response: This is quiet a constructive opinion which is important for our research. Thank you very much for the comments and it has been added in the text.
4.8 Figure 5. "Velocity" .
Response: Sorry for the mistaken. The misspelling of the name word on the ordinate of Figure 5 has been corrected.
4.9 L211-225 Support discussion using literature.
Response: We agree. We have used the literature discussion.
4.10 L256-265 Support discussion using literature.
Response: We agree. We have used the literature discussion.
4.11 L252-272 Important explanation, however, should be made in the methodology.
Response: We agree. We have described equations 9 and 10 in the Materials and methods.
4.12 L342 That is why it is important to describe the characteristics of the plug in the methodology.
Response: We agree. We have described the characteristic parameters of the plug in Methods.
Reviewer 3 Report
Figures 1 and 6 could be improved. Suggested comments are provided in the attached file.

Author Response
Figures 1 and 6 could be improved. Suggested comments are provided in the attached file.
Response 1: We agree. We have merged Figure 6 into Figure 1. At the same time, we agree with the comments in the attached file, which have been revised as indicated in the file.
Round 2
Reviewer 2 Report
The manuscript makes a significant contribution to furrow irrigation studies. The author has included many suggestions. The manuscript can be published.